# Comparison Between Non-Invasive Methane Measurement Techniques in Cattle

**DOI:** 10.3390/ani9080563

**Published:** 2019-08-15

**Authors:** Jagoba Rey, Raquel Atxaerandio, Roberto Ruiz, Eva Ugarte, Oscar González-Recio, Aser Garcia-Rodriguez, Idoia Goiri

**Affiliations:** 1Department of Animal Production, NEIKER—Tecnalia, Granja Modelo de Arkaute, Apdo. 46, 01080 Vitoria-Gasteiz, Spain; 2Departamento de Producción Agraria, Escuela Técnica Superior de Ingeniería Agronómica, Alimentaria y de Biosistemas, Universidad Politécnica de Madrid, Ciudad Universitaria s/n, 28040 Madrid, Spain; 3Departamento de Mejora Genética Animal, Instituto Nacional de Investigación y Tecnología Agraria y Alimentaria, 28040 Madrid, Spain

**Keywords:** NDIR, laser, agreement, enteric emissions, interchangeability

## Abstract

**Simple Summary:**

Enteric methane emissions pose a serious issue to ruminant production and environmental sustainability. To mitigate methane emissions, combined research efforts have been put into animal handling, feeding and genetic improvement strategies. For all research efforts, it is necessary to record methane emissions from individual cows on a large scale under farming conditions. The objective of this trial was to compare two large-scale, non-invasive methods of measuring methane (non-dispersive infrared methane analyzer (NDIR) and laser), in order to see if they can be used interchangeably. For this, paired measurements were taken with both devices on a herd of dairy cows and compared. Significant sources of disagreement were identified between the methods, such that it would not be possible to use both methods interchangeably without first correcting the sources of disagreement.

**Abstract:**

The aim of this trial was to study the agreement between the non-dispersive infrared methane analyzer (NDIR) method and the hand held laser methane detector (LMD). Methane (CH_4_) was measured simultaneously with the two devices totaling 164 paired measurements. The repeatability of the CH_4_ concentration was greater with the NDIR (0.42) than for the LMD (0.23). However, for the number of peaks, repeatability of the LMD was greater (0.20 vs. 0.14, respectively). Correlation was moderately high and positive for CH_4_ concentration (0.73 and 0.74, respectively) and number of peaks (0.72 and 0.72, respectively), and the repeated measures correlation and the individual-level correlation were high (0.98 and 0.94, respectively). A moderate concordance correlation coefficient was observed for the CH_4_ concentration (0.62) and for the number of peaks (0.66). A moderate-high coefficient of individual agreement for the CH_4_ concentration (0.83) and the number of peaks (0.77) were observed. However, CH_4_ concentrations population means and all variance components differed between instruments. In conclusion, methane concentration measurements obtained by means of NDIR and LMD cannot be used interchangeably. The joint use of both methods could be considered for genetic selection purposes or for mitigation strategies only if sources of disagreement, which result in different between-subject and within-subject variabilities, are identified and corrected for.

## 1. Introduction

The livestock sector plays an important role in climate change, representing 14.5% of all anthropogenic emissions. Within livestock, one of the most relevant sources of greenhouse gases is enteric methane (CH_4_) representing 39% of sector emissions [1]. Methane is a potent greenhouse gas with a high global warming potential, calculated as 25 times greater than that of carbon dioxide (CO_2_) [2]. For these reasons, enteric CH_4_ has become a great concern worldwide. In fact, agreements like “The Paris Agreement” in 2015 have pointed out the importance of the reduction of CH_4_ emission by cattle.

Moreover, enteric CH_4_ production is associated with considerable dietary energy losses for ruminants, ranging from 1.7% to 14.9% of gross energy intake for lactating cows [3] that could lead to decreased energy gain and productivity. Feed is typically the main production cost item in mixed and intensive systems. Wasting part of the feed energy in the form of CH_4_ is not only a climate change issue but also a production problem. In this context, the reduction of livestock greenhouse gases emissions, and in particular enteric CH_4_, is posed as a top issue in the agribusiness sector. 

The reduction of these emissions can be addressed from the combination of handling, feeding and genetic improvement strategies. However, in order to implement any of these strategies and especially to breed animals with lower CH_4_ production, it is necessary to have individual CH_4_ data on a large number of animals in commercial conditions. 

Methane concentration is heritable [4], and accurate and reliable individual measurements are necessary to include this trait in the breeding programs. These measurements must be made on a large number of animals under commercial conditions [5]. Several methods to record CH_4_ emissions from individual animals have been proposed, each with specific advantages, scope of application and also flaws [6,7,8]. Rapid and non-labor-extensive methods of measuring CH_4_ are required in order to implement national genetic evaluations. The non–dispersive infrared analyzer CH_4_ analyzer (NDIR) and the hand-held laser CH_4_ detector (LMD) are two alternative methodologies that measure CH_4_ concentrations in the breath of dairy cattle. Both technologies are non-invasive and allow high throughput in commercial conditions. Both methods have been proved to reliably quantify individually CH_4_ concentrations in exhaled air at farm conditions [9,10,11,12].

Garnsworthy et al. [13] used for the first time a NDIR CH_4_ detector to quantify CH_4_ emissions from individual cows on a farm, by sampling air released by eructation during milking. With this technique, a sampling point was installed inside the feed bin of automated milking systems (AMS) and CH_4_ concentration on exhaled air was continuously measured in each cow’s visits to the AMS. The LMD is a hand-held gas detector for remote measurements of column density for CH_4_ containing gases that was originally developed for the detection of gas leaks. However, Chagunda, et al. [14] demonstrated its ability to quantify CH_4_ concentration in exhaled air and estimate enteric CH_4_ output in dairy cows. Their conclusions were later corroborated by other authors [15,16].

Generating large databases of CH_4_ emissions that help addressing reduction strategies has been lately proposed. Exchanging data from different measurements devices in different countries for more accurate studies or genetic evaluations would be an advantage. Thus, harmonizing measurement methodologies is a main concern in recent years, and assessing the equivalence or the statistical agreement between measuring methods is crucial to propose strategies to combine data from different devices. It is important to establish the agreement prior to measuring large numbers of individuals. One approach to analyze this agreement, usually used in medical field, is to record simultaneous repeated records on different subjects using the two methods [17] in order to analyze different sources of disagreement like differences on population means, on between-subject or on within-subject variabilities, as well as the calculation of agreement indices, such as concordance correlation coefficient (CCC) and the coefficient of individual agreement (CIA).

Up to date, few comparative studies have been conducted on these two instruments to determine their equivalence or lack thereof [18], but no study has compared their agreement on paired measurements on the same breath air samples, up to the best of our knowledge. Therefore, the aim of this study was to analyze the agreement and concordance of CH_4_ emissions through LMD and the NDIR on paired measurements under field conditions.

## 2. Materials and Methods 

### 2.1. Experimental Setup and CH_4_ Concentration Determination

The LMD is a hand held open path laser measuring device. The model used in this study was LaserMethaneMini (Tokyo Gas Engineering Co., Ltd. Anritsu Devices Co., Ltd., Tokyo, Japan). The principle of the LMD measuring technology was described previously by Chagunda et al. [12,13,14]. Briefly, this device is based on infrared absorption spectroscopy using a semiconductor laser for CH_4_ detection. The device must be pointed towards the nostrils of the cow from a fixed distance. Then, the LMD measures the density of the air column between the device and the animal’s nostrils. The reflected laser beam is detected by the device, and its signal is processed and converted to the cumulative CH_4_ concentration along the laser path in ppm-m. The LMD was connected to a tablet (Samsung Galaxy Tab A6, New Jersey, USA) running GasViewer app (Tokyo Gas Engineering Solutions, Tokyo, Japan) via Bluetooth connection for exporting and storing the data in real time at 0.5 s intervals. The effect of atmospheric ambient CH_4_ concentration from the measurements was discounted using the offset function of the LMD.

The NDIR (Guardian NG Edinburg Instruments Ltd., Livinstong, UK) is one of the so-called sniffer methods that measure CH_4_ concentration (ppm) in breath or exhaled air. These methods have been previously used by Garnsworthy et al. [10] to assess the CH_4_ production of dairy cows at commercial farms. Briefly, a gas sampling tube from the front of a cow’s head to a gas analyzer to continuously measure CH_4_ concentration in the cow’s breath is used. Then, air is drawn through the instrument by an integral pump between the gas inlet port and analyzer. The device used in this study had a range of 0 to 10,000 ppm. For this study, air was sampled continuously at a rate of 1 L/min through an 8 mm polyamide tube, using approximately 2 m of tube from the analyzer to cow´s nostrils. Methane concentration was recorded at 1 s intervals and stored in a datalogger (Data Recorder SRD-99; Simex Sp. z o.o, Gdańsk, Poland). Baseline or ambient CH_4_ concentration was calculated as mean CH_4_ concentration before starting the measurements and subtracted from the measured data. Each day before starting measurements, the NDIR analyzer was verified using standard mixtures of CH_4_ in nitrogen (0.0%, 0.25%, 0.50%, 0.75% and 1.0%; MESA International Technologies INC, Santa Ana, CA, USA). 

For this trial, records were measured in 29 Holstein (11) and Brown Swiss (18) dairy cows in the Fraisoro Agricultural School (Zizurkil, Spain). All animal experiments were carried out in accordance with EU Directive 2010/63/EU for animal experiments and approved by the ethics committee (NEIKER-OEBA-2017-004). Cows were 18.5% of the 1st parity, 52% of the 2nd parity, 18.5% of the 3rd parity and 11% of the 4th parity. Average days in milk at the beginning of the experiment was 102 ± 88 d. Cows were offered a partial mixed ration consisting of corn silage, grass silage and straw ad libitum, and a mean of 5 kg of concentrate supplied in the AMS.

Breath CH_4_ concentration was measured on six different days during four months, except for two cows that died during the experimental period. Measurements were performed after morning unifeed distribution between 10:00 and 14:00. Animals were restrained and CH_4_ was measured simultaneously with the two devices during a 5 min sampling period, obtaining a total of 164 paired measurements. An operator pointed the LMD at a cow’s nostril at a fixed distance of 1 m and trying to maintain the angle from which the LMD was pointed to the cow. Data was recorded every 0.5 s in a tablet. Another operator simultaneously placed NDIR sampling tube on the cow’s nostrils in order to measure CH_4_ concentration in exhaled air with the NDIR and data was recorded every 1 s in a datalogger.

### 2.2. Calculations and Statistical Analysis

The average CH_4_ concentration from LMD and NDIR was calculated as the arithmetic mean of all concentration values within each 5 min profile measured with the LMD and the NDIR analyzer respectively. The number of eructation events or peaks in each 5 min profile was also calculated for both devices as described in [18].

In order to perform a concordance analysis, phenotypes with the same units are required. Some assumptions must be taken into account in order to convert mean LMD values in ppm-m into ppm. Data obtained with LMD at a distance of one meter (D) was transformed considering an exhaled air bubble of 10 cm (X) from the cow’s nostrils, and 2 ppm CH_4_ concentration in the environment air (A). The resulting phenotype was calculated as:CH_4_(ppm-m) = X(m) × CH_4_(ppm) + D(m) × A(ppm)(1)

Data was analyzed using a linear mixed effects model with Kronecker product covariance structure in a doubly multivariate set-up [19] using the MIXED procedure of SAS with longitudinal repeated measures (SAS^®^ Institute INC, Cary, NC, EEUU, Version 7.15, 2017). A Kenward-Roger correction was used to compute the correct denominator degrees of freedom of fixed effects in the presence of repeated measures [20]. The linear mixed effects model for the ith records can be written as:(2)$$y_{i}=\;µ+b_{1}\left(S_{i1}\right)+b_{2}\left(S_{i2}\right)+b_{3}D+b_{4}B+b_{i1}\left(Z_{i1}\right)+b_{i2}\left(Z_{i2}\right)+e_{i}$$
where every method replication response is denoted by superscript i and y_i_ is the method replication response (CH_4_ concentration, ppm). Terms S_i1_ and S_i2_ indicate the instrument each response belongs to (LMD and NDIR) and will take a value of either zero or one to link y_i_ to the corresponding instrument. The regression coefficients b_1_ and b_2_ are the respective fixed effects of instruments, b_3_ denotes the fixed effect of day of measurement and b_5_ is the fixed effect of the breed. Terms b_i1_ and b_i2_ are random effect parameters for each cow with each of the two devices, with bi ~ ND (0,G), and Z_i1_ and Z_i2_ relate responses y_i_ to the respective methods. Term e_i_ is the random residuals from each instrument response with e_i_ ~ ND (0,R_i_).

The method described in detail by Roy [19] was employed to formally test the significance of differences for the between-subject and within-subject variances of the two instruments by means of a log likelihood ratio test between models differing for various combinations of structured and unstructured variance–covariance matrices G and R_i_. Using the variance components of the previous model, CIA and CCC were calculated.

Repeatability for each method was calculated as the between-cow $\left(\sigma_{B}^{2}\right)$ variance divided by the between-cow and within-cow ($\sigma_{W}^{2})\;$ variance.

(3)$$\mathrm{REP}=\frac{\sigma_{B}^{2}}{\sigma_{B}^{2}+\sigma_{W}^{2}}$$

Additionally, the repeated measures correlation (r_p_) was calculated as:(4)$$r_{p}=\frac{\sigma_{B1B2}^{2}}{\sqrt{\sigma_{B1}^{2}\;\times {\;\sigma }_{B2}^{2}}}$$
with σ_B1B2_^2^ being the covariance between cows for the LMD and NDIR, and σ_B1_^2^ and σ_B2_^2^ being the between-subject variance for method 1 and 2, respectively.

Individual level of correlation was calculated using variance components applying the statistical model described previously but including the stage of lactation and lactation number as fixed effects [21].

A Pearson correlation coefficient and Spearman´s rank correlation test were used to assess the association between the overall mean CH_4_ concentrations across all paired individual cow records [22]. 

## 3. Results and Discussion

Strategies to address CH_4_ abatement strategies need large amounts of CH_4_ measurements. Combining measurements from different countries has been proposed as a possible strategy and is being studied elsewhere. However, different research groups often utilize different types of device to record CH_4_. In this context, a common goal is to decide if the measurement systems agree suitably to each other, and thus can be used interchangeably. The current study is the first directly comparing spot-sampling methods based on breath analysis called the NDIR and LMD simultaneously in the same breath air samples.

An example of a CH_4_ profile of a cow as measured with the NDIR sensor and the LMD can be seen in Figure 1. Five different eructation events or CH_4_ peaks can be seen with both devices. Mean number of peaks differed between instruments (Table 1), being the LMD capable of detecting higher number of peaks (4.69 vs. 4.24, *p* < 0.001), this could be related with the higher sensitivity of the LMD to detect small variations on CH_4_ concentration, even away from the animal [11] and with the time of recording (every 0.5 s vs. 1 s with LMD and NDIR, respectively). However, these differences on the mean number of peaks are of small biological relevance. Moreover, the between subjects variance for the number of peaks did not differ between instruments and a large and positive repeated measures correlation and individual-level correlation were observed (Table 1). Thus, it can be said that both methods are capable to detect the eructation events or peaks in a similar way.

In our study, a mean CH_4_ concentration of 97 ppm-m with a minimum of 21 ppm-m and a maximum of 303 ppm-m were recorded with the LMD. With the NDIR sensor a mean CH_4_ concentration of 1268 ppm with a minimum of 58 ppm and a maximum of 3575 ppm was recorded. The values observed with the LMD agree with those observed in other studies; for instance, Chagunda et al. [12,13,14] found average CH_4_ concentrations between 107–369 ppm-m, depending on the animal activity. The values observed with the NDIR sensor are in line with those observed in other studies [23].

Before considering any method for comparison one needs to ensure that its replication error is acceptable. Repeatability is relevant to the study of method comparison because the repeatabilities of the two methods limit the amount of agreement, which is possible. If one method has poor repeatability the agreement between the two methods is likely to be poor too. As shown in Table 1, the repeatability for the concentration of CH_4_ was greater with the NDIR sensor (0.42) than with the LMD (0.23). However, for the number of peaks, the repeatability with the LMD (0.20) was greater than with the NDIR sensor (0.14). The repeatability values obtained in the current trial with the NDIR sensor agree with those reported by Lassen et al. [24] made in the AMS, and with Negussie et al. [25] who reported a repeatability value of 0.36, measuring CH_4_ based on CH_4_/CO_2_ ratio, but were lower than those reported by Sorg et al. [18] who obtained a repeatability of 0.77 (0.13). However, the latter authors processed the NDIR data using head lifting algorithms and modeled it in a Fourier series approach on the time of day of measurement to account for diurnal patterns of CH_4_ emission as described by Difford et al. [26]. Repeatability from measurements obtained with LMD was lower than that obtained with the NDIR but similar to those found by other authors [18,27]. The LMD measures were less repeatable than those of NDIR probably because the LMD measurement took place in the open space of the barn, while the NDIR sampled air in the nostrils where environmental influences are likely less variable, e.g., air movement due to ventilation and wind speed.

Though discouraged in comparison studies with repeated measures per subject, we provide the Pearson’s and Spearman’s correlation coefficients, which were moderately high and positive between instruments for CH_4_ concentration (0.73 and 0.74, respectively) and the number of peaks (0.72 and 0.72, respectively). These correlation coefficients could be shrunk because repeated measurements per subject inflate the residual error variance. In these cases, the repeated measures correlation is more appropriate because the effect of repeated measurements per subject are accounted for and prevents a downward bias of correlations due to imprecise measurements. In this sense, repeated measures correlation and individual-level correlation between methods were high (0.98 and 0.94, respectively) for CH_4_ concentration (Table 1), which was higher than that observed by Sorg et al. [18]. Individual-level correlations have been used as proxies for genetic correlations in difficult or expensive to measure traits, being the genetic correlation the most informative correlation metric for assessing how to incorporate a method in a selection index [21].

We did not observe a significant breed effect. The CH_4_ concentrations population means and all variance components (between-subject and within-subject) differed (*p* < 0.05) between instruments (Table 1). It must be pointed out that this study aimed to assess the agreement or lack thereof between LMD and NDIR. Neither methods are considered as the gold standard, but they allow high throughput recording in a whole cattle population, which is of main interest for the dairy industry. This method comparison study aimed to evaluate whether these methods can be used interchangeably in commercial farms. Traditionally, if there is no reference method, assessing agreement has been based on the intraclass correlation coefficient and CCC. In this study a moderate CCC was observed between instruments for both CH_4_ concentration (0.62) and for the number of peaks (0.66; Table 1). This CCC value was higher than that observed by Sorg et al. [18] probably due to the high variance estimates in the present study. This coefficient with fixed within-subject variability increase as between-subject variability increases and some authors question whether this coefficient is adequate in assessing agreement at the individual level, being necessary to evaluate agreement using both CCC and CIA in order to overcome the inherent shortcomings of both metrics [28,29]. The CIA evaluates agreement relative to imprecision in such a way that a good individual agreement means that the individual difference between readings from different methods is close to the difference between replicated readings within a method, and could therefore be inflated by low repeatability values. We observed higher values of CIA between instruments for both CH_4_ concentration (0.83) and number of peaks (0.77) than CCC values, which can be explained by the observed moderate repeatability values (Table 1). However, CIA values were moderately high and based on the threshold of “good” interchangeability (0.455) defined by Barnhart et al. [17], CH_4_ concentration measurements of both instruments might be used interchangeably.

Nevertheless, based on the criteria for the statistical agreement of the equivalence of means and total variability suggested by Roy [19], the observed differences in the mean populations, between-subject variabilities and within-subject variabilities (Table 1), CH_4_ concentration obtained by means of NDIR and LMD could not be considered interchangeable. When two methods are compared without a gold standard, neither provides an unequivocally correct measurement, but CH_4_ measures obtained with both devices can be used interchangeably provided that the sources of disagreement are identified and corrected. In this sense, Difford et al. [26] in a study comparing two sniffer methods with simultaneous repeated measurements proposed a process of calibration and standardization to overcome these sources of disagreement with positive results.

## 4. Conclusions

Methane concentration measurements obtained by means of NDIR and LMD instruments cannot be directly used interchangeably under commercial conditions. The joint use of both methods could be considered to establish classifications of individuals, in relation to their CH_4_ emissions, in studies for genetic selection purposes or to evaluate CH_4_ emissions reduction strategies only if sources of disagreement, which result in different between-subject and within-subject variabilities, are identified and corrected for.

## Figures and Tables

**Figure 1 animals-09-00563-f001:**
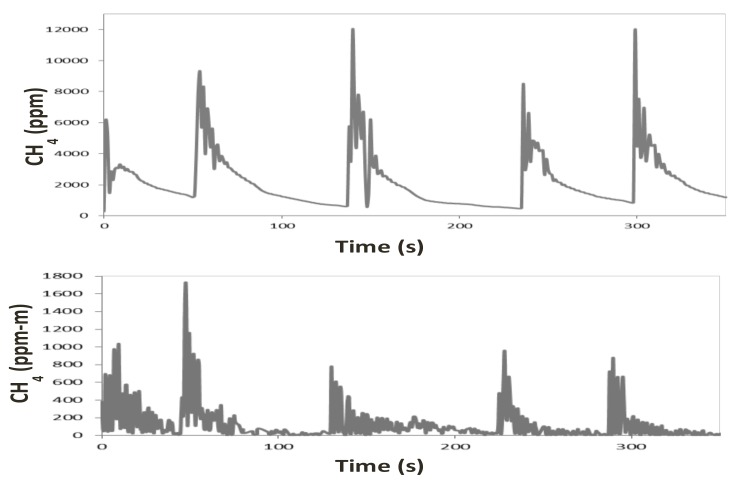
Representation of CH_4_ concentration for a cow measured with a non-dispersive infrared (NDIR) sensor (above), and laser methane detector (LMD; below) with different belching events.

**Table 1 animals-09-00563-t001:** Sources of (dis)agreement for the CH_4_ concentration and number of peaks.

Item	CH_4_ Concentration in (ppm)		Number of Peaks	
	NDIR	LMD_cal	NDIR	LMD_cal
LSMean (SE)	1280 (88.3) ^a^	991 (57.3) ^b^	4.24 (0.1) ^a^	4.69 (0.1) ^b^
Between-cow variation	149081 ^a^	54556 ^b^	0.18	0.19
Within-cow variation	352398 ^a^	187742 ^b^	1.16 ^a^	0.76 ^b^
Repeatability	0.42	0.23	0.14	0.20
Repeat measures correlation	0.98		1.00	
Individual level correlation	0.94		1.00	
Pearson correlation	0.73		0.72	
Spearman correlation	0.74		0.72	
Concordance correlation coefficient	0.62		0.66	
Coefficient of individual agreement	0.83		0.77	

NDIR: Non-dispersive infrared CH_4_ sensor; LMD_cal: Laser CH_4_ detector transformed to ppm; SE: Standard error; estimates with subscripts differ (*p* < 0.05).

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
