# Peer review of "Comparison Between Non-Invasive Methane Measurement Techniques in Cattle"

_animals, 2019, doi:10.3390/ani9080563_

Round 1

Reviewer 1 Report

The present study is focused on comparison between non-invasive methods for methane measurements. This is an important area of research considering other methods (chambers, Greenfeed system) are very expensive and cannot be used elsewhere. The objectives are clear and manuscript is written well which is easy to follow. Following are my comments:

-       My major concern with this paper is lack gold standard for methane emissions. I do not see the use of comparing both methods if none of the methods evaluated are precise and the only way to measure accuracy is to compare them with gold standards.  

-       No information was provided on the calibration of both instruments.

-       Methane was measured for 6 days during 4 months. What was the frequency of measurement? Why did authors decide to measure only for 6 days when cows were used for 4 months? Why authors decided to collect data only for 4 hours (10:00-14:00) when repeated measurements over entire day could have generated diurnal variation in methane emissions.

-       Did authors measure individual cow intake? Comparing both instruments on the intensity of methane production (adjusted for dry matter intake) would be very useful.

-       Please explain what is ppm-m and how equations 1 and 2 were derived.

-       Figure 1. Please provide values for methane in ppm for LDR.

-       L225: Please discuss values in ppm

Author Response

We really appreciate your comments to improve the paper. I hope all your questions have been adressed. See our response in the attached document

Reviewer 2 Report

Review Animal

Dear Editors, I have read the works of Rey et al., with great interest. This is one of the few instrument/method comparisons I have read, where the design is statistically very strong and purely for the purpose of instrument comparisons. So often animal scientists in the disciplines of nutrition complicate their comparisons with the added effects of different diets and geneticists are solely interested in genetics correlations and estimated breeding values and fail to give the phenotype due diligence. The authors of this manuscript have not fallen into either trap. Although these two methods have been compared under commercial conditions previously, the measurements were not made simultaneously and therefore some doubt remained. I believe Rey et al., by measuring cows with both methods simultaneously have put all doubts about the agreement between these two methods to rest. The design, results and interpretation are sound, however the text can be difficult to read and I have some comments for the authors to address, which I believe will strengthen their work. I have tried to assist the authors with English usage in the important sections ‘Simple Summary’, ‘Abstract’ and ‘Conclusions’ but recommended that the authors have a native English speaker go over the entire manuscript to ensure their good work is a pleasure to read. I want to encourage the authors, they have done a very good job of reconciling instrument comparisons with genetics and with some added analyses this will be a great paper.

Major comments:

I would like to help clarify some concepts to the authors here, the statistical wording used across the different measurement comparison disciplines are very different and the assumptions under which correlations hold can be very different. Pearson’s and spearman’s correlations are discouraged in measurement comparisons studies because measurement comparison studies make use of repeated measures per subject(cow). This inflates the residual error variance and as a result shrinks the Pearson or Spearman’s correlations (It is good the authors showed these correlations). In these cases, the repeated measures correlation is more appropriate because the effect of repeated measurements per subject are accounted for and prevents downward bias of correlations due to measurement imprecision. The calculation of CCC and CIA requires a correlation (in the numerator for the CCC) and (in the denominator for the CIA). In measurement comparisons studies, the correlation coefficient which goes into the CCC and CIA should be the repeated measures correlation because we need repeated measures per cow. I have checked your calculations of CH4 concentration in Table 1 for CCC and CIA you are correct.

 However, it is assumed that you are measuring a single population, not two populations or in your case breeds. If the effect of breed is significant, the repeated measures correlation will be inflated, and so too will the CCC and CIA. Therefore, you need to take the effect of breed into account in your calculation of repeated measures correlation, CCC and CIA. If my suspicions are incorrect, then there will be no changes, but if you indeed have a heterogenous population, then the repeated measures correlation, CCC and CIA will shrink. This must be done.

If you take this a step further and include effects in your model like, parity or lactation number, lactation stage, diet (if there are different diets) and other non-genetic effects (if they are contributing to between subject variability), then you will get individual level correlations. These are very useful because they are proxies for genetic correlations. If two methods have individual level correlations different from zero, then both methods can still be used for genetic improvement by making a selection index even if the CCC and CIA are low. These concepts have always been clearly discussed across the studies you cite e.g Sorg et al., 2018, Difford et al., 2016 because these studies used a single breed, or individual level correlations were not the focus.

See (Dingemanse, N. J., & Dochtermann, N. A. (2013)) for help with different within and between subject variances and correlations and Difford et al., (2018) for an example of individual level correlations used in a heterogenous population instrument comparison as a proxy for genetic correlations.

In summary:

1.)   Include breed in your model and recalculate repeated measures correlation, CCC and CIA.

2.)   Include lactation number, lactation stage and/or other non-genetics effects such as diet if there are any, in a second model and calculate individual level correlations.

Minor comments about scientific content:

Lines 39-42. I agree with the last line of the abstract. But some important distinctions need to be made inorder to be fully correct. ‘Both methods can only be used for measuring methane concentrations of dairy cows only if the sources of disagreement are identified and corrected for.’ This is true at the phenotype level. Then for genetic selection there are slightly different requirements ‘Both methods could be used for genetic selection provided they are heritable and sufficiently genetically corrected for inclusion in a selection index’ – I say this because you do not estimated genetic correlations, you estimate repeated measures correlations. Your repeated measures correlations are likely a biased over representation of the genetic correlation between the methods because you do not fully account for non-genetic sources of between cow variability such as the effect of breed and lactation number as well as lactation stage (I see measurements were over 4 months). I recommend you estimate individual level correlations as well, these could be used as proxies for genetic correlations and would then allow you to make a stronger final sentence in your abstract about genetic selection.

Line 51 – provide the full name of the ‘Paris’ agreement for the reader. ‘The Paris Agreement’

Lines 66. You cite two heritability papers for methane production. Yet, you do not measure methane production, instead measure methane concentration. There are other papers like ‘van Engelen et al., 2018’ who report methane concentration is heritable as well.

Lines 66-69. I am very certain Kirsty Hammond has not said anything to back this statement up in her GreenFeed paper.

Line 113. This sentence reads as if ethical clearance is needed for measuring using the LMD. But the LMD is meant to be non-invasive. I think this ethical statement belongs elsewhere in the manuscript and probably pertains to all animal testing in the trial and not only the LMD. Perhaps lines 142-147.

Can you provide summary information about your cows, numbers per breed, lactation number, stage etc?

Were there different diets in this study, and what was the diet – Just feed ingredients is fine. Were cows grazing? Fed partial or total mixed rations, were they fed concentrates?

Lines 175 – 201. Here I think you need to add a second model, which will include all the effects of the current model, but will also include effect of breed and lactation number. Then the sample calculation can be done for the repeated measures correlation, but not you will have an individual-level correlation which will add to your manuscript, because it is one step closer to genetic correlations.

Lines 202 – why is SAS referenced here? You already mentioned SAS on lines 173.

Lines 203 – 209 appear to be discussion and do not fit with the start of the results section.

Lines 159-201. Peak counting is not described in the materials and methods.

Lines 257 – 259. Bear in mind that repeated measures correlation is based on between cow variation, Sorg et al measured in Holstein cows, your repeated measures correlation is higher, but it could be due to you having two breeds. It OK for you to make this comparison, but I want you to calculate individual level correlations as well so that we can also see if the effect of breed is inflating your repeated measures correlation. See major comments.

Line 260-264. Good to see this sentence – I agree.

Line 272-276. Also a good statement. But remember you might have inflated between subject variability due to breed.

Lines 283 – 294. I agree, this is a down side of the CIA when methods have high within subject variability. Usually, the CCC will capture this and be low, but if the repeated measures correlation is inflated (as I suspect ours is because of breed or some other effect inflating between cow variability), then CCC and CIA will both be high (and misleading), luckily the added criteria from Roy still capture these sources of disagreement. I suspect that your repeated measures correlation, CCC and CIA will all come down if breed or some other effect is inflating between subject variation, then all methods will likely agree that these two instruments are not interchangeable. 

Line 297-299. An added point here about standardisation and calibration, you can only do this when you have simultaneous repeated measurements, (As you do in this study).   

Conclusion. If you provide individual level correlations, then you can probably conclude more about genetic selection.

Minor comments about English grammar spelling and verbiage (Many of them):

I have many comments about the simple summary – this paragraph has been simplified and reduced to the point that the context is lost for most sentences it is difficult to know what the authors are referring to. Comments below:

Lines 13-14. Cumbersome sentence, plus missing a frame of reference, is this about cows or ruminants or transport or crop production? – suggest changing to ‘Enteric methane emissions pose a serious issue to ruminant production and environmental sustainability’

Lines 14-13. Cumbersome sentence – suggest changing to ‘To mitigate methane emissions, combined research efforts have been put into animal handling, feeding and genetic improvement strategies.’

Lines 15-16. Suggest changing to ‘For all research efforts, it is necessary record methane emissions from individual cows on a large scale under farming conditions’ 

Lines 17 – 18. Objective is lost in the wording. Suggest changing to: ‘The objective of this trial is to compare two large-scale, non-invasive methods of measuring methane (NDIR and Laser), in order to see if they can be used interchangeably.’

Lines 20 – 21. Suggest ‘Paired measurements were taken with both devices on a herd of dairy cows and compared.’

Lines 22- 23. I don’t think it is necessary to prequalify your conclusion with ‘With all the results obtained’, we do not expect you to make conclusions based on anything else. Suggest changing sentence to ‘Significant sources of disagreement were identified between the methods, such that it would not be possible to use both methods interchangeably without first correcting the sources of disagreement.’

Line 26. It is confusing that you define the abbreviation for methane (CH4) within the same of one instrument ‘infrared methane (CH4) analyzer method’ Remove this abbreviation here and instead define it in line 27.

Lines 28-29. Suggest splitting this sentence and changing to ‘The Repeatability of CH4 concentration was greater with the NDIR 29 (0.31) than for the LMD (0.25).’

Lines 29-30. ‘For the number of peaks, repeatability of the LMD was greater (0.22 vs. 0.17, respectively).’

Line 32. Insert the word ‘the’ – ‘… and the repeated measures correlation…’

Line 38-39. Remove ‘data sets’ and change to ‘records’ or ‘phenotypes’ or ‘measurements’.

Lines 56 – 60. Two very indirect and difficult sentences full of commas – suggest rewording.

Lines 66-88. This part of the introduction is full of long, compounded sentences – please reword in simple shorter sentences.

Line 63. ‘specially’ change to ‘especially’ or ‘particularly’.

Lines 66. Simplify ‘Methane production is an inheritable character’ to ‘Methane production is heritable’

Line 131 – subscript on CH4

Lines 240 – 247. This is a very long sentence and difficult to read.

Line 301. Same comment  Line 38-39. Remove ‘data sets’ and change to ‘records’ or ‘phenotypes’ or ‘measurements’.

Table 1.  ‘Repetibility’ – ‘Repeatability’

Table 1. Add individual level correlations

Literature Cited by Reviewer:

Van Engelen, S., Bovenhuis, H., van der Tol, P. P. J., & Visker, M. H. P. W. (2018). Genetic background of methane emission by Dutch Holstein Friesian cows measured with infrared sensors in automatic milking systems. Journal of dairy science101(3), 2226-2234.

Dingemanse, N. J., & Dochtermann, N. A. (2013). Quantifying individual variation in behaviour: mixedeffect modelling approaches. Journal of Animal Ecology82(1), 39-54.

Difford, G. F., Olijhoek, D. W., Hellwing, A. L. F., Lund, P., Bjerring, M. A., de Haas, Y., ... Løvendahl, P. (2019). Ranking cows’ methane emissions under commercial conditions with sniffers versus respiration chambers. Acta Agriculturae Scandinavica A: Animal Sciences68(1), 25-32. https://doi.org/10.1080/09064702.2019.1572784

Author Response

We really appreciate your comments to improve the paper. They have really been helpful. I hope all your questions have been adressed. See our response in the attached document

Round 2

Reviewer 1 Report

Dear authors,

Thanks for addressing my comments. 

Author Response

Dear reviewer

Thanks a lot for your valuable comments to improve the article. We really appreciate your time and efforts.